# On the Correspondence between Subshifts of Finite Type and Statistical Mechanics Models

**DOI:** 10.3390/e24121772

**Published:** 2022-12-03

**Authors:** Luis Armando Corona, Raúl Salgado García, Edgardo Ugalde

**Affiliations:** 1Instituto de Investigación en Ciencias Básicas y Aplicadas, Universidad Autónoma del Estado de Morelos, Cuernavaca 62209, Mexico; 2Centro de Investigación en Ciencias-IICBA, Universidad Autónoma del Estado de Morelos, Cuernavaca 62209, Mexico; 3Instituto de Física, Universidad Autónoma de San Luis Potosí, San Luis Potosi 78250, Mexico

**Keywords:** subshifts of finite type, statistical mechanics models, phase transitions, 37B10, 82B20

## Abstract

Several classical problems in symbolic dynamics concern the characterization of the simplex of measures of maximal entropy. For subshifts of finite type in higher dimensions, methods of statistical mechanics are ideal for dealing with these problems. R. Burton and J. Steif developed a strategy to construct examples of strongly irreducible subshifts of finite type admitting several measures of maximal entropy. This strategy exploits a correspondence between equilibrium statistical mechanics and symbolic dynamics—a correspondence which was later formalized by O. Häggström. In this paper, we revisit and discuss this correspondence with the aim of presenting a simplified version of it and present some applications of rigorous results concerning the Potts model and the six-vertex model to symbolic dynamics, illustrating in this way the possibilities of this correspondence.

## 1. Introduction

A subshift of finite type is a symbolic dynamical system determined by a finite collection of forbidden patterns. For transitive one-dimensional subshifts of finite type on a finite number of states, there exists one and only one invariant measure achieving the topological entropy, which, on the other hand, is the supremum of the metric entropies (see [1] for instance). A topological dynamical system with a unique measure of maximal entropy is qualified as intrinsically ergodic. In higher dimensions, transitivity is not enough to ensure that a subshift of finite type is intrinsically ergodic. In [2,3] R. Burton and J. Steif developed a strategy to construct examples of transitive subshifts of finite type admitting several measures of maximal entropy. This strategy, further developed by O. Häggström in [4,5], consists on making subshifts of finite type correspond to statistical mechanics models, in such a way that equilibrium states for the statistical mechanics model correspond to measures of maximal entropy for the symbolic system. The success of this approach lies in the fact that it furnishes a dictionary between equilibrium statistical mechanics and symbolic dynamics, translating rigorous results from statistical mechanics to symbolic dynamics. Indeed, one of the results obtained in [2], the existence of a strongly irreducible subshift of finite type in dimension, with two supporting at least two ergodic measures of maximal entropy, is the translation of a result by Peierls concerning the Ising model. Using the same strategy, Burton and Steif derive in [3], using an idea analogous to the one used by M. Zahradnik in [6], a complete description of the simplex of measures of maximal entropy. In [5], Häggström formalizes and generalizes the above-mentioned correspondence in such a way that for each equilibrium state of the statistical mechanics model (SMM), there is a measure of maximal entropy for the corresponding subshift of finite type. In this paper, we revisit this correspondence with the aim of pointing out further applications of statistical mechanics results to symbolic dynamics. The correspondence we study in this paper is a simplified version of the one due to O. Häggström appearing in [5]. Ours is simpler in what concerns the construction the subshift of finite type, as well in what concerns the proof of the equivalence. Our construction also makes explicit the correspondence between the parametrization of the family of SFTs and the inverse temperature in the corresponding SMM, making sense of a phase transition in the symbolic context. Although the construction can be carried out in any dimension, for the sake of concreteness, we will restrict ourselves to dimension two, where the relevant phenomenology already appears. In order to illustrate the aforementioned correspondence and the nature of the applications we invoke, we use the Potts model and the six-vertex model of statistical mechanics. The rest of the paper is organized as follows: in Section 2, we introduce some basic notions that we use throughout the paper. In Section 3, we develop the SFT–SMM correspondence. In Section 4 and Section 5, we illustrate this correspondence in the case of the Potts and the six-vertex model, respectively. We finish with some concluding remarks.

## 2. Definitions and Notations

We place ourselves in the common setting of 2D symbolic dynamics and 2D lattice statistical mechanics. Configurations are $Z^{2}$-arrays with entries in a finite set A (the set of occupation numbers, spins, energy levels, etc.), which we call the alphabet. To denote the coordinate projections of a configuration *x*, we use subindices. Hence, for $z\in Z^{2}$, with $x_{z}$ we denote the projection of *x* on the coordinate *z*. Similarly, if $\Lambda \subset Z^{2}$, then $x_{\Lambda }\in A^{\Lambda }$ denotes the patch in Λ obtained from *x* by coordinate projections. To signify that $\Lambda \subset Z^{2}$ is finite, we use the notation $\Lambda \in Z^{2}$. Now, for each $\Lambda \in Z^{2}$, every Λ-shaped patch $a\in A^{\Lambda }$ defines a cylinder set
$$\left[a\right]:=\left\{x:\;x_{\Lambda }=a\right\}.$$ We supply the space of configurations with the distance $d\left(x,y\right)=e^{-\min\{n\geq 0:\;|z|\leq n\Rightarrow x_{z}=y_{z}\}}$. Each $z\in Z^{2}$ defines a shift transformation $\sigma_{z}$ such that ${\left(\sigma_{z}x\right)}_{s}=x_{z+s}$ for each configuration *x* and every $s\in Z^{2}$. The group $\sigma :=\{\sigma_{z}:\;z\in Z^{2}\}$ of shfit transformations acts continuously on the space of configurations. A set *X* of configurations is a subshift if it is closed with respect to the distance *d* and σ-invariant (σ-invariant means that $\sigma_{z}X=X$ for all $z\in Z^{2}$). The term subshift is used to refer to the metric space as well as to the dynamical system defined on it by the action of σ. A subshift is of finite type if for some $\Lambda \subset Z^{2}$ and a finite collection of Λ-shaped patches $L\subset A^{\Lambda }$ we have
$$X:=\left\{x:\;{\left(\sigma_{z}x\right)}_{\Lambda }\in L\;\;\mathrm{for}\;\mathrm{all}\;z\in Z^{2}\right\}.$$ Subshifts are equivalently defined by a collection of forbidden patches. The patch $a\in A^{\Lambda }$ is *X*-admissible (admissible for short) if there is at least one configuration in *X* containing that patch, i.e., [a]∩X≠∅. We denote with $L_{\Lambda }\left(X\right)$ the collection of all the admissible Λ-shaped patches. Whenever there is no ambiguity, we use [a] as a shorthand for [a]∩X. We also refer to *X* as the subshift, understanding that it is subject to the action of σ.

Let $\Lambda ,\Lambda^{\prime }\subset Z^{2}$ be disjoint. For each $a\in A^{\Lambda }$ and $b\in A^{\Lambda^{\prime }}$, with a⊕b, we denote the $\Lambda \cup \Lambda^{\prime }$-shaped patch $c\in A^{\Lambda \cup \Lambda^{\prime }}$ such that $c_{\Lambda }=a$ and $c_{\Lambda^{\prime }}=b$. The subshift (X,σ) is strongly irreducible if there exists ℓ>0 such that for each couple of disjoint shapes $\Lambda^{\prime },\Lambda ⋐Z^{d}$ with $\mathrm{dist}(\Lambda ,\Lambda^{\prime })\geq \ell$ and, every couple of admissible patches $a\in A^{\Lambda },\;b\in A^{\Lambda^{\prime }}$, there is an admissible configuration containing both patches, i.e., [a⊕b]≠∅.

Let us assume that (X,σ) is strongly irreducible. For each n∈N, let $\Lambda_{n}=\left[-n,n\right]\times \left[-n,n\right]\cap Z^{2}$. The topological entropy of the subshift $X\subset A^{Z^{2}}$ is the limit
$$h_{\mathrm{top}}\left(X\right):=\lim_{n\to \infty }\frac{\log\left|L_{\Lambda_{n}}\left(X\right)\right|}{|\Lambda_{n}|}.$$ The existence of this limit follows from the Fekete’s subadditivity lemma. The topological entropy of *X* gives the rate of exponential growth of $|L_{\Lambda }\left(X\right)|$ with respect to |Λ|.

The collection of all the Borel probability measures on *X* is denoted by $M^{1}\left(X\right)$. It is a convex set, and it is a topological space by considering the weak topology. The subcollection of σ-invariant Borel probability measures, which we denote $M_{\sigma }^{1}\left(X\right)$, is a simplex in this topology. A measure $\mu \in M_{\sigma }^{1}\left(X\right)$ is ergodic if any σ-invariant set has a μ measure equal to zero or one. The measure theoretic entropy for $\mu \in M_{\sigma }^{1}\left(X\right)$ is the quantity
$$h\left(\mu \right):=-\lim_{n\to \infty }\frac{1}{|\Lambda_{n}|}\sum_{a\in A^{\Lambda_{n}}}\mu \left[a\right]\log\mu \left[a\right].$$ Once again, the existence of the limit is consequence of the subadditivity lemma.

We consider two kinds of simplices of measures on $M_{\sigma }^{1}\left(X\right)$: the equilibrium states for an interaction at a given inverse temperature, and the simplex of measures of maximal entropy. For this, we need to remind readers of some notions related to statistical mechanics. We start by considering an interaction, which is a collection of functions $\Phi :=\left\{\Phi_{\Lambda }:A^{\Lambda }\to R:\;\Lambda ⋐Z^{2}\right\}$. In the following, we assume that the interaction is of finite range and σ-invariant. This means that
(a)There exists r>0 such that if diam(Λ)>r then $\Phi_{\Lambda }\left(a\right)=0$ for all $a\in A^{\Lambda }$, and(b)$\Phi_{\Lambda }\left(a\right)=\Phi_{\Lambda +z}\left(\sigma_{z}a\right)$, for each $z\in Z^{2}$.

Abusing notation, we use $\sigma_{z}a$ to denote the unique patch in $b\in A^{\Lambda +z}$ such that $b_{s}=a_{s-z}$, for all $s\in Z^{2}$. The number *r* in (a) stands for the range of the interaction. For each $\Lambda ⋐Z^{2}$ let $\partial^{r}\Lambda$ be the *r*-border of Λ, i.e., $\partial^{r}\Lambda :=\left\{z\in Z^{2}\backslash \Lambda :\mathrm{dist}\left(z,\Lambda \right)\leq r\right\}$. In the particular case of r=1, we use the notation ∂Λ instead of $\partial^{1}\Lambda$.

Given the interaction Φ, an equilibrium state at inverse temperature β≥0 is a probability measure $\mu \in M_{\sigma }^{1}\left(X\right)$ satisfying the following: for each $\Lambda ⋐Z^{2}$, $\Lambda^{\prime }⊃\Lambda \cup \partial^{r}\Lambda$ and every admissible patch $a\in A^{\Lambda^{\prime }}$, we have
$$\mu \left(\left[a_{\Lambda }\right]|a\right):=\frac{e^{-\beta H_{\Lambda }\left(a_{\Lambda \cup \partial^{r}\Lambda }\right)}}{\sum_{b\in A^{\Lambda }:\;\left[b⊕a_{\partial^{r}\Lambda }\right]\neq \emptyset }e^{-\beta \;H_{\Lambda }\left(b⊕a_{\partial^{r}\Lambda }\right)}},$$
where $H_{\Lambda }\left(y\right):=\sum_{U\subset \Lambda \cup \partial^{r}\Lambda }\Phi_{U}\left(y_{U}\right)$ is the energy of the configuration *y* restricted to the volume Λ. The collection of all the equilibrium states, $E_{\beta }\left(\Phi \right)\subset M_{\sigma }^{1}\left(X\right)$, is a Choquet simplex whose extrema are ergodic measures (see [7] for instance). The denominator
$$Z_{\beta }\left(\Lambda ,a_{\partial^{r}\Lambda }\right):=\sum_{b\in A^{\Lambda }:\;\left[b⊕a_{\partial^{r}\Lambda }\right]\neq \emptyset }e^{-\beta \;H_{\Lambda }\left(b⊕a_{\partial^{r}\Lambda }\right)},$$
defines the partition function, which depends on the inverse temperature β as well as the boundary patch $a_{\partial \Lambda }\in A^{\partial^{r}\Lambda }$. If the (X,σ) is strongly irreducible, the limit
$$f_{\phi }\left(\beta \right):=-\frac{1}{\beta }\;\lim_{n\to \infty }\frac{1}{|\Lambda_{n}|}\log\;Z_{\beta }\left(\Lambda_{n},x\right),$$
exists and defines the Helmholtz free energy.

From the finite range σ-invariant interaction Φ, we define the specific energy $u\left(x\right):=\sum_{U∋o}\Phi_{U}\left(x_{U}\right)/\left|U\right|$, where o denotes the origin of $Z^{2}$, i.e., o=(0,0). If (X,σ) is a strongly irreducible SFT, for each $\mu \in M_{\sigma }^{1}\left(X\right)$, we have
$$f_{\phi }\left(\beta \right)\leq \mu \left(u\right)-\frac{h_{\sigma }\left(\mu \right)}{\beta }.$$ The equality is attained if and only if $\mu \in E_{\beta }\left(\Phi \right)$. Since $\lim_{\beta \to 0}\beta \;f\left(\beta \right)=-h_{\mathrm{top}}\left(X\right)$, then $h_{\sigma }\left(\mu \right)\leq h_{\mathrm{top}}\left(X\right)$ for each $\mu \in M_{\sigma }^{1}\left(X\right)$. The collection $\mathrm{Max}\left(X\right):=\{\mu \in M_{\sigma }^{1}\left(X\right):\;h_{\sigma }\left(\mu \right)=h_{\mathrm{top}}\left(X\right)\}$ is the simplex of measures of maximal entropy, and it coincides with $E_{0}\left(\Phi \right)$ for an arbitrary interaction Φ, as long as the local energy $x\mapsto u\left(x\right):=\sum_{o\in \Lambda }\Phi_{\Lambda }\left(x\right)$ is a continuous function. As mentioned before, when Max(X) is a singleton, we say that (X,σ) is intrinsically ergodic.

We can transform any finite range σ-invariant interactions over an arbitrary SFT, by means of a block coding, to a one-letter function over an SFT defined by a collection of patches on the cross-like volume
$$\Lambda_{\mathrm{cross}}:=\left\{\left(0,0\right),\left(\pm 1,0\right),\left(0,\pm 1\right)\right\}\equiv \left\{o,\pm e^{1},\pm e^{2}\right\}\subset Z^{2}.$$ For this, let *Y* be a two-dimensional SFT on the alphabet B, defined by the collection of *F*-shaped admissible patches. Let Ψ be an interaction of finite range. Consider the volume
$$\Lambda =F⋃\left(\underset{U∋o:\;\Psi_{U}\neq 0}{⋃}U\right),$$
which comprises the range of the interaction as well as the volume needed to define the SFT. We can naturally embed *Y* into a two-dimensional subshift ι(Y):=X on the alphabet $A:=B^{\Lambda }$. The embedding y↦ι(y) is given by $\iota {\left(y\right)}_{z}=y_{z+\Lambda }$ for all $z\in Z^{2}$, and it is a topological conjugacy between (Y,σ) and (X,σ). Clearly, *X* is the SFT defined by the collection
$$L:=\left\{a\in A^{\Lambda_{\mathrm{cross}}}:a_{z}\in L_{\Lambda }\left(Y\right)\;\mathrm{and}\;{\left(a_{o}\right)}_{\zeta +z}={\left(a_{z}\right)}_{\zeta }\;\forall z\in \Lambda_{\mathrm{cross}}\;\forall \zeta \in \Lambda \;s.t.\;\left\{\zeta +z,z\right\}\subset \Lambda \right\}.$$ This is nothing but the requirement of the letters of an admissible patch in *X* to be admissible patches in *Y*, and that they correctly overlap when considered as patches in *Y*. The embedding ι reduces Ψ to the one-letter function $a\mapsto \Phi_{\left\{o\right\}}\left(a\right)=\sum_{U∋o}\Psi_{U}\left(a_{U}\right)/\left|U\right|$ which defines the energy functions $a\mapsto H_{\Lambda }\left(a\right)=\sum_{z\in \Lambda }\Phi_{\left\{o\right\}}\left(b_{z}\right)$. The embedding ι induces the map $\iota^{*}:M_{\sigma }^{1}\left(Y\right)\to M_{\sigma }^{1}\left(X\right)$ such that $\iota^{*}\mu \left(B\right)=\mu \left(\iota^{-1}B\right)$ for each Borel set B⊂X. This map is an isomorphism between the simplices $M_{\sigma }^{1}\left(Y\right)$ and $M_{\sigma }^{1}\left(X\right)$ mapping $E_{\beta }\left(\Psi \right)$ into $E_{\beta }\left(\Phi \right)$ for each β≥0.

## 3. The SFT–SMM Correspondence

As mentioned in the introduction, the Häggström correspondence takes advantage of the fact that for some models of equilibrium statistical mechanics, the energy is concentrated on some lattice regions that we refer to as contours. In the complement of these regions, the configurations have to be homogeneous and of minimal energy. The passage from one thermodynamic regime, where only one equilibrium state exists, to the coexistence of several ergodic equilibrium states is governed by the competition between energy and entropy. The corresponding Burton–Steif family of subshifts replaces homogeneous regions by highly entropic regions, while contours remain zero-entropy or low-entropy regions. The transition is then driven by the increase in entropy of the homogeneous regions. Let us now present a simplified version of the Häggstrom construction, which makes explicit the correspondence between a parametrized family of SFTs and an SMM subject to the variation of the inverse temperature. The construction can be carried out in any dimension, but for concreteness, we restrict it to dimension two, where phase transitions may occur.

The framework is that of finite-range interactions on strongly irreducible SFTs in dimension two. Taking into account the observations at the end of the previous section, we can assume that the set of admissible patches have support in the cross-shaped region, $\Lambda_{\mathrm{cross}}:=\left\{o,\pm e^{1},\pm e^{2}\right\}\subset Z^{2}$, and that $\Phi_{\Lambda }=0$ if |Λ|≠1. Since Φ is σ-invariant, then $\Phi_{\left\{z\right\}}\left(a\right)=\Phi_{\left\{o\right\}}\left(\sigma_{-z}a\right)$ for all $z\in Z^{2}$. In order to establish the equivalence between equilibrium states and measures of maximal entropy, we make the following assumption concerning the interaction.

**Hypothesis** **1.**
*There exists $\epsilon_{0}>0$ and $S⋐N_{0}$ such that $\Phi_{\left\{o\right\}}\in \epsilon_{0}\;S$ for each $\Lambda ⋐Z^{2}$.*


Therefore, we assume that all the values of the interaction are multiples of the same magnitude $\epsilon_{0}$. This is in fact equivalent to the restriction of an interaction taking only rational values.

Let us partition the alphabet A into equi-energetic letters, i.e., for each s∈S let
$$A_{s}:=\left\{a\in A:\;\Phi_{\left\{o\right\}}\left(a\right)=\epsilon_{0}\;s\right\}.$$ Let us now split the alphabet in order to define a parametrized family of SFTs, with each value of the parameter corresponding to a different inverse temperature for the corresponding SMM. For each $N\in N_{0}$ let $A_{N}:={⋃}_{s\in S}\left(A_{s}\times \left\{0,1,\ldots ,N_{s}-1\right\}\right)$ with $N_{s}:=N^{\max S-s}$. Hence, the least possible degeneracy corresponds with the maximum local energy while degeneracy is a monotonous function of the energy. We can think of the first coordinate in $A_{s}\times \left\{0,1,\ldots ,N_{s}-1\right\}$ as the color type of the symbol, while the second coordinate will be the tone of the corresponding color. The split alphabet $A_{N}$ has as many color types as A, but the *s*-th color type is split into $N_{s}$ possible tones. We use $\pi_{c}$ and $\pi_{t}$ for the projection onto the color and the tone, respectively. We use the same notation, $\pi_{c}$ and $\pi_{t}$, for the extension of these projections to finite patches and infinite configurations. For the letters in the split alphabet, we use boldface font to distinguish them from the letters in the original alphabet.

Let us now define the two-dimensional SFT $X_{N}$ on the alphabet $A_{N}$, whose collection of admissible patches is
$$L_{N}:=\left\{a\in A_{N}^{\Lambda_{\mathrm{cross}}}:\;\pi_{c}\left(a\right)\in L\right\}.$$ It is easily verified that $\left(X_{N},\sigma \right)$ inherits the strong irreducibility from (X,σ). The projection $\pi_{c}:X_{N}\to X$ is a factor map (continuous, commuting with σ) and the induced transformation $M_{\sigma }^{1}\left(X_{N}\right)∋\mu \mapsto \pi^{*}\mu =\mu \circ \pi_{c}^{-1}$ is continuous with respect to the weak topologies, and it is linear.

The main result, which is analogous and to some extent summarizes Theorems 4.1 and 4.2 in [5], is the following.

**Theorem** **1.***Let X be a two-dimensional SFT and* Φ *a one-letter interaction. For each N∈N let $\beta_{N}=\log\left(N\right)/\epsilon_{0}$. The transformation $\mu \mapsto \pi_{c}^{*}\left(\mu \right):=\mu \circ \pi_{c}^{-1}$ is an homeomorphism between $\mathrm{Max}(X_{N})$ and $E_{\Phi }\left(\beta_{N}\right)$ with respect to the respective weak topologies, and it is such that $\pi_{c}^{*}\left(\lambda \mu +\left(1-\lambda \right)\nu \right)=\lambda \;\pi_{c}^{*}\mu +\left(1-\lambda \right)\;\pi_{c}^{*}\nu$ for each $\mu ,\nu \in \mathrm{Max}(X_{N})$ and every λ∈[0,1].*

Notice that $\pi_{c}^{*}$ establishes a one-to-one correspondence between ergodic measures in $\mathrm{Max}(X_{N})$ and $E_{\beta_{N}}\left(\Phi \right)$. It is this fact that allows the construction of strongly irreducible SFTs which are not intrinsically ergodic.

We also have the following.

**Proposition** **1.**
*Let f(β) be the Helmholtz free energy at inverse temperature β≥0 and for each N∈N, let $\beta_{N}=\log\left(N\right)/\epsilon_{0}$. Then, $h_{\mathrm{top}}\left(X_{N}\right)=\log\left(N\right)\;\max S-\beta_{N}\;f\left(\beta_{N}\right)$.*


The proof of Theorem 1 follows from some classic results in Statistical Mechanics and direct computations. All the ideas behind the proof already appear in the works by Burton and Steif [2,3] and the subsequent works by Häggström [4,5]. Nevertheless, in the version presented here, the proof is greatly simplified, underlining the key ideas.

We require some additional notation. For each $\Lambda ⋐Z^{2}$, $a\in A^{\Lambda }$ and $x\in A_{N}^{\partial \Lambda }$, let
(1)$$\begin{array}{ccc} \Omega_{\Lambda }\left(x\right) & := & \left\{b\in A_{N}^{\Lambda }:\left[b⊕x\right]\neq \emptyset \right\}, \end{array}$$
(2)$$\begin{array}{ccc} \Omega_{\Lambda }\left(a,x\right) & := & \left\{a\in \pi_{c}^{-1}\left(a\right):\;\left[a⊕x\right]\neq \emptyset \right\}. \end{array}$$
We have the following.

**Lemma** **1.**
*For N∈N let $\beta_{N}=\log\left(N\right)/\epsilon_{0}$, and let $\mu_{N}\in \mathrm{Max}\left(X_{N}\right)$. For each $\Lambda ⋐Z^{2}$, $a\in A_{N}^{\Lambda }$ and $x\in A_{N}^{\partial \Lambda }$, we have*

(3)
$$\mu_{N}\left(\left[a\right]|x\right)=\frac{\mu_{\beta_{N}}\left(\left[a\right]|x\right)}{|\Omega_{\Lambda }\left(a,x\right)|}=\frac{e^{\beta_{N}\;H_{\Lambda }\left(a⊕x\right)}}{e^{\beta_{N}\;\left(\epsilon_{0}\max S\right)\;\left|\Lambda \right|}},$$

*where $a=\pi_{c}\left(a\right)$ and $x=\pi_{c}\left(x\right)$.*


**Proof.** Let us start by noticing that [a⊕x]≠∅ if and only if [a⊕x]≠∅. Let us assume that [a⊕x]≠∅, otherwise the equalities (Equation 3) trivially hold. Since $\mathrm{Max}(X_{N})$ and $E_{0}\left(\Psi \right)$ coincide for an arbitrary continuous interaction Ψ, the volume-Λ conditional measure $\mu_{N}\left(•\;|x\right)$, of any $\mu_{N}\in \mathrm{Max}\left(X_{N}\right)$, is necessarily uniformly distributed on the set $\Omega_{\Lambda }\left(x\right)$, i.e.,
(4)$$\mu_{N}\left(\left[a\right]|x\right):=\frac{1}{|\Omega_{\Lambda }\left(x\right)|}.$$We associate with each $a\in A^{\Lambda }$ and s∈S the level-set
$$\gamma_{\Lambda }^{s}\left(a\right):=\left\{z\in \Lambda :\;a_{z}\in A_{s}\right\}.$$ All the patches in $\Omega_{\Lambda }\left(a,x\right)$ have the color type of *a*, while the tone at each site can take any of the values compatible with the corresponding color type; therefore,
(5)$$\Omega_{\Lambda }\left(a,x\right)=\prod_{s\in S}\prod_{z\in \gamma_{\Lambda }^{s}\left(a\right)}\left(\left\{a_{z}\right\}\times \left\{0,1,\ldots ,N_{s}-1\right\}\right).$$
Let us remind readers that $N_{s}=N^{\max S-s}$. Taking this into account, by using (Equation 5) and the equipartition property of $\mu_{N}$, we obtain
(6)$$\begin{array}{ccc} |\Omega_{\Lambda }\left(a,x\right)| & = & \prod_{s\in S}\prod_{z\in \gamma_{\Lambda }^{s}\left(a\right)}\;N_{s}=e^{\sum_{s\in S}\left|\gamma_{\Lambda }^{s}\left(a\right)\right|\;\log\left(N_{s}\right)}=N^{|\Lambda |\;\max S}\left(e^{-\log\left(N\right)\sum_{s\in S}s\;\left|\gamma_{\Lambda }^{s}\left(a\right)\right|}\right), \end{array}$$
(7)$$\begin{array}{ccc} |\Omega_{\Lambda }\left(x\right)| & = & \sum_{b\in A^{\Lambda }}\left|\Omega_{\Lambda }\left(b,x\right)\right|=\sum_{[b⊕x]\neq \emptyset }\prod_{s\in S}\prod_{z\in \gamma_{\Lambda }^{s}\left(b\right)}\;N_{s}\\ & = & N^{|\Lambda |\;\max S}\sum_{\sum_{[b⊕x]\neq \emptyset }}e^{-\log\left(N\right)\sum_{s\in S}s\;\left|\gamma_{\Lambda }^{s}\left(b\right)\right|}. \end{array}$$ Hence,
$$\begin{array}{cc} \mu_{N}\left(\pi_{c}^{-1}\left[a\right]|x\right) & \equiv \frac{|\Omega_{\Lambda }\left(a,x\right)|}{|\Omega_{\Lambda }\left(x\right)|}=\frac{e^{-\log\left(N\right)\sum_{s\in S}s\;\left|\gamma_{\Lambda }^{s}\left(a\right)\right|}}{\sum_{b\in A^{\Lambda }:\;\left[b⊕x\right]\neq \emptyset }e^{-\log\left(N\right)\sum_{s\in S}s\;\left|\gamma_{\Lambda }^{s}\left(b\right)\right|}}\\ \; & =\frac{e^{-\beta_{N}\;\epsilon_{0}\sum_{s\in S}s\;\left|\gamma_{\Lambda }^{s}\left(a\right)\right|}}{\sum_{b\in A^{\Lambda }:\;\left[b⊕x\right]\neq \emptyset }e^{-\beta_{N}\;\epsilon_{0}\sum_{s\in S}s\;\left|\gamma_{\Lambda }^{s}\left(b\right)\right|}}=\frac{e^{-\beta_{N}\;H_{\Lambda }\left(a⊕x\right)}}{\sum_{b\in A^{\Lambda }}e^{-\beta_{N}\;H_{\Lambda }\left(b⊕x\right)}}=\mu_{\beta_{N}}\left(\left[a\right]|x\right). \end{array}$$ Notice that $H_{\Lambda }\left(b⊕x\right)$ depends on *x* only due to the fact that b⊕x has to be an admissible patch. Finally, since $\mu_{N}\left(•|x\right)$ is the uniform measure on $\Omega_{\Lambda }\left(x\right)$, then
$$\mu_{N}\left(\left[a\right]|x\right)=\frac{\mu_{\beta_{N}}\left(\left[a\right]|x\right)}{|\Omega_{\Lambda }\left(a,x\right)|}=\mu_{\beta_{N}}\left(\left[a\right]|x\right)\frac{e^{\beta_{N}\;H_{\Lambda }\left(a⊕x\right)}}{e^{\beta_{N}\;\left(\epsilon_{0}\max S\right)\;\left|\Lambda \right|}}.$$□

### 3.1. Proof of Theorem 1

**Proof.** Let us start by noticing that $\mathrm{Max}(X_{N})$ coincides with the simplex of equilibrium states on $X_{N}$, for an energy identical to zero. A direct computation allows us to verify that $\pi_{c}^{*}$ is linear in the space of signed measures, and therefore it is such that
$$\pi_{c}^{*}\left(\lambda \mu +\left(1-\lambda \right)\nu \right)=\lambda \;\pi_{c}^{*}\mu +\left(1-\lambda \right)\;\pi_{c}^{*}\nu ,$$
for each $\mu ,\nu \in \mathrm{Max}(X_{M})$ and every λ∈[0,1]. It is also easy to verify that $\pi_{c}^{*}$ is continuous with respect to the weak topologies. Let us take $\mu_{N}\in \mathrm{Max}\left(X_{N}\right)$ and let $\nu =\pi_{c}^{*}\left(\mu_{N}\right)$. Since $\mu_{N}$ is a Markov field, then for each $\Lambda ⋐Z^{2}$, $a\in Z_{q}^{\Lambda }$ and $x\in Z_{q}^{\partial^{\Lambda }}$, we have
$$\nu \left(\left[a\right]|x\right):=\frac{\nu ([a⊕x])}{\nu ([x])}=\frac{\mu_{N}\left(\left[\pi_{c}^{-1}\left(a⊕x\right)\right]\right)}{\mu_{N}\left(\left[\pi_{c}^{-1}\left(x\right)\right]\right)}=\frac{\sum_{x\in \pi_{c}^{-1}\left(x\right)}\mu_{N}\left[x\right]\mu_{N}\left(\pi_{c}^{-1}\left[a\right]|x\right)}{\sum_{x\in \pi_{c}^{-1}\left(x\right)}\mu_{N}\left[x\right]}.$$ According to Lemma 1,
$$\mu_{N}\left(\pi_{c}^{-1}\left[a\right]|x\right)=\sum_{a\in \pi^{-1}\left(a\right)}\mu_{N}\left(\left[a\right]|x\right)=\mu_{\beta_{N}}\left(\left[a\right]|x\right),$$
therefore,
$$\nu \left(\left[a\right]|x\right)=\mu_{\beta_{N}}\left(\left[a\right]|x\right)\;\frac{\sum_{x\in \pi_{c}^{-1}\left(x\right)}\mu_{N}\left[x\right]}{\sum_{x\in \pi_{c}^{-1}\left(x\right)}\mu_{N}\left[x\right]}=\mu_{\beta_{N}}\left(\left[a\right]|x\right),$$
which proves that $\pi_{c}^{*}\left(\mu_{N}\right)\in E\left(\beta_{N}\right)$.

Now, for $\mu_{\beta_{N}}\in E\left(\beta_{N}\right)$, let $\nu \in M(X_{N})$ be such that
(8)$$\nu \left(\left[a\right]\right):=\frac{\mu_{\beta_{N}}\left(\left[\pi_{c}\left(a\right)\right]\right)}{|\pi_{c}^{-1}\left\{\pi_{c}\left(a\right)\right\}|},$$
for each $\Lambda ⋐Z^{2}$ and every $a\in A_{N}^{\Lambda }$. Clearly, ν is a probability measure, and since $\mu_{\beta_{N}}$ is σ-invariant, then for each $z\in Z^{2}$, we have
$$\nu \left(\sigma_{z}\left[a\right]\right)=\frac{\mu_{\beta_{N}}\left(\sigma_{z}\left[\pi_{c}\left(a\right)\right]\right)}{|\pi_{c}^{-1}\left\{\pi_{c}\left(a^{\prime }\right)\right\}|}=\frac{\mu_{\beta_{N}}\left(\left[\pi_{c}\left(a\right)\right]\right)}{|\pi_{c}^{-1}\left\{\pi_{c}\left(a\right)\right\}|}=\nu \left(\sigma_{z}\left[a\right]\right),$$
where $a^{\prime }\in A_{N}^{\Lambda -z}$ is such that $a_{s}=a_{s+z}^{\prime }$ for each s∈Λ. Therefore, ν is σ-invariant as well. On the other hand, for each $x\in A_{N}^{\partial \Lambda }$, and taking into account Lemma 1, we have
$$\begin{array}{cc} \nu ([a]|x)\; & :=\frac{\nu ([a⊕x])}{\nu ([x])}=\frac{\mu_{\beta_{N}}\left(\left[\pi_{c}\left(a⊕x\right)\right]\right)}{\mu_{\beta_{N}}\left(\left[\pi_{c}\left(x\right)\right]\right)}\;\frac{|\pi_{c}^{-1}\left\{\pi_{c}\left(x\right)\right\}|}{|\pi_{c}^{-1}\left\{\pi_{c}\left(a⊕x\right)\right\}|}\\ \; & =\frac{\mu_{\beta_{N}}\left(\left[\pi_{c}\left(a\right)\right]|\pi_{c}\left(x\right)\right])}{|\Omega_{\Lambda }\left(a,x\right)|}\;\frac{|\Omega_{\Lambda }\left(a,x\right)|\;|\pi_{c}^{-1}\left\{\pi_{c}\left(x\right)\right\}|}{|\pi_{c}^{-1}\left\{\pi_{c}\left(a⊕x\right)\right\}|}\\ \; & =\mu_{N}\left(\left[a\right]|x\right)\;\frac{|\Omega_{\Lambda }\left(a,x\right)|\;|\pi_{c}^{-1}\left\{\pi_{c}\left(x\right)\right\}|}{|\pi_{c}^{-1}\left\{\pi_{c}\left(a⊕x\right)\right\}|}, \end{array}$$
where $\mu_{N}\left(\left[a\right]|x\right)$ is the common value of the conditional probability for any measure $\mu_{N}\in \mathrm{Max}\left(X_{N}\right)$. One can easily verify that
$$|\Omega_{\Lambda }\left(a,x\right)|=\frac{|\pi_{c}^{-1}\left\{a⊕x\right\}|}{|\pi_{c}^{-1}\left\{x\right\}|},$$
therefore, $\nu \left(\left[a\right]|x\right)=\mu_{N}\left(\left[a\right]|x\right)$. Hence, $\nu \in \mathrm{Max}(X_{N})$, and since $\nu \circ \pi_{c}^{-1}\left(\left[a\right]\right)=\mu_{\beta_{N}}\left(\left[a\right]\right)$, then the transformation $\mu_{\beta_{N}}\mapsto \nu$ defined by (Equation 8) gives the inverse of $\pi_{c}^{*}$. The continuity of this transformation with respect to the weak topologies is a direct consequence of its very definition, and its verification is straightforward.

From all the above, it follows that $\pi_{c}^{*}$ is a bicontinuous linear bijection between the simplices $\mathrm{Max}(X_{N})$ and $E(\beta_{N})$. □

### 3.2. Proof of Proposition 1

**Proof.** According to (Equation 7), for each $\Lambda ⋐Z^{2}$ and $x\in A_{N}^{\partial \Lambda }$, we have
$$\begin{array}{cc} |\Omega_{\Lambda }\left(x\right)|=\sum_{b\in A^{\Lambda }}\left|\Omega_{\Lambda }\left(b,x\right)\right|\; & =e^{\left(\log(N)\;\max S)|\Lambda \right|}\sum_{\sum_{[b⊕x]\neq \emptyset }}e^{-\beta_{N}\;H_{\Lambda }\left(b⊕x\right)}\\ \; & \equiv e^{\left(\log(N)\;\max S)|\Lambda \right|}Z_{\beta_{N}}\left(\Lambda ,x\right), \end{array}$$
where $x=\pi_{c}\left(x\right)\in A^{\partial \Lambda }$ and $Z_{\beta_{N}}\left(\Lambda ,x\right)$ is the partition function. Since
$$\left|\Omega_{\Lambda }\left(x\right)\right|\leq \left|\{b\in A_{N}^{\Lambda }:\;\left[b\right]\neq \emptyset \}\right|\leq \sum_{y\in A_{N}^{\partial \Lambda }}\left|\Omega_{\Lambda }\left(y\right)\right|\leq e^{\left(\log|A|+\log(N)\;\max S)|\partial \Lambda \right|}\;\left|\Omega_{\Lambda }\left(x\right)\right|,$$
then,
(9)$$\begin{array}{ccc} \frac{\log\left(\left|\{b\in A_{N}^{\Lambda }:\;\left[b\right]\neq \emptyset \}\right|\right)}{\left|\Lambda \right|}\; & \geq & \beta_{N}\;\epsilon_{0}\;\max S+\frac{\log Z_{\beta_{N}}\left(\Lambda ,x\right)}{\left|\Lambda \right|}\\ \; & \leq & \beta_{N}\;\epsilon_{0}\;\max S+\frac{\log Z_{\beta_{N}}\left(\Lambda ,x\right)}{\left|\Lambda \right|}\\ \; & \; & \;+\beta_{N}\;\epsilon_{0}\max S\frac{\left|\partial \Lambda \right|}{\left|\Lambda \right|}. \end{array}$$
The Helmholtz free energy is at inverse temperature $\beta_{N}$ the limit
$$f\left(\beta_{N}\right)=-\frac{1}{\beta_{N}}\lim_{n\to \infty }\frac{\log Z_{\beta_{N}}\left(\Lambda_{n},x\right)}{|\Lambda_{n}|}.$$
Hence, by inequalities (Equation 9) we finally obtain
(10)$$\begin{array}{ccc} h_{\mathrm{top}}\left(X_{N}\right) & := & \lim_{n\to \infty }\frac{\log\left(\left|\{b\in A_{N}^{\Lambda_{n}}:\;\left[b\right]\neq \emptyset \}\right|\right)}{|\Lambda_{n}|} \end{array}$$
(11)$$\begin{array}{ccc} & = & \beta_{N}\left(\epsilon_{0}\max S-f\left(\beta_{N}\right)\right)=\log\left(N\right)\max S-\beta_{N}\;f\left(\beta_{N}\right). \end{array}$$□

Theorem 1 allows us to exhibit transitive SFTs having simplices of maximizing measures with a particular structure, for instance, equal to the standard (q−1)-simplex in dimension *q*, as we show in the next section. Furthermore, if for an SMM of the kind considered here we are able to compute the Helmholtz free energy, then Proposition 1 gives us a family of SFTs for which the exact value of the topological entropy can be explicitly given. In the next section, we illustrate these applications using the Potts model.

## 4. The SFT-SMM Correspondence for the Potts Model

The two-dimensional version of the Potts model was introduced by Potts in [8], generalizing a method to find the critical temperature introduced some years before, in the context of the two-dimensional Ising model, by Kramers and Wannier [9,10]. In this model, the underlying subshift is $\left(X:={\left\{0,1,\ldots ,q-1\right\}}^{Z^{2}},\sigma \right)$, the integer *q* being the number of colors. The interaction Φ is such that
$$\Phi_{\Lambda }\left(a\right)=\left\{\begin{array}{cc} \delta (a_{z},a_{z^{\prime }}) & \;\mathrm{if}\;\Lambda =\left\{z,z^{\prime }\right\}\;\mathrm{and}\;\left|z-z^{\prime }\right|=1,\\ 0 & \;\mathrm{otherwise}, \end{array}\right.$$
with δ(·,·) the Kronecker’s delta. We can identify the Potts model with a strongly irreducible SFT *X* on the alphabet $A={\left\{0,1,\ldots ,q-1\right\}}^{\Lambda_{\mathrm{cross}}}$. The set of admissible patches $L\subset A^{\Lambda_{\mathrm{cross}}}$ defining *X* is determined by the overlapping of symbols in A when considered as patches in ${\left\{0,1,\ldots ,q-1\right\}}^{\Lambda_{\mathrm{cross}}}$. After this identification, the energy of volume-Λ is determined by the one-letter interaction
$$a\mapsto \Phi_{\left\{o\right\}}\left(a\right)=\frac{1}{2}|\{z\in \Lambda :\;|z|=1\;\mathrm{and}\;\;a_{z}=a_{o}\}|.$$
This interaction satisfies Hypothesis 1 with S={0,1,2,3,4} and $\epsilon_{0}=1/2$.

The phase diagram of the Potts model, i.e., the complete description of the simplex of equilibrium states E(β) for all $\beta \in R^{+}$, was described by Martirosian in [11]. There, he proves that for a sufficiently large *q* (according to Baxter [12], a large *q* would mean q>4, which was recently proved by Duminil-Copin and coauthors [13]), there is a critical inverse temperature $\beta_{c}$ such that E(β) has exactly q+1 extremal measures for $\beta =\beta_{c}$, it has exactly *q* extremal measures for $\beta >\beta_{c}$, and it is a singleton if $\beta <\beta_{c}$. For the sake of completeness, let us state a version of Martirosian’s theorem, with the improvements by Duminil-Copin et al. [13,14], adapted to our needs.

**Theorem** **2.**
*Let $\beta_{c}\left(q\right):=\log\left(\sqrt{q}+1\right)/2$ for each q≥2.*

*Let q∈{2,3,4}. For $\beta \in [0,\beta_{c}\left(q\right)]$, there is a unique equilibrium state; meanwhile, for $\beta >\beta_{c}\left(q\right)$, there are exactly q different ergodic equilibrium states.*

*Let q≥5. For $\beta \in [0,\beta_{c}\left(q\right))$, there is a unique equilibrium state; for $\beta =\beta_{c}\left(q\right)$, there are exactly q+1 different ergodic equilibrium states; and for $\beta >\beta_{c}\left(q\right)$, there are exactly q different ergodic equilibrium states.*


This result is derived from the aforementioned works by Martirosian and Duminil-Copin and coauthors. The recent results by Duminil-Copin and coauthors improved Martirosian’s theorem, among other things, since they rigorously established the critical number of colors, q=5, from which the phase transition is discontinuous. Furthermore, they prove that transition at $\beta_{c}\left(q\right)$ is sharp, which means that the influence of the boundary conditions decays exponentially quickly below the critical inverse temperature. Above the critical inverse temperature, a unique color k∈{0,1,…,q−1} corresponds to each ergodic equilibrium state which, with a probability of one, fills an unbounded connected component of $Z^{2}$, while the rest of the colors occupy only bounded connected components or islands.

The critical inverse temperature, $\beta_{c}\left(q\right):=\log\left(\sqrt{q}+1\right)/2$, was obtained from a duality argument first introduced by Kramers and Wannier. It was proved to be exact for q=2 from Onsager’s result [15]. For q≥4, the conjectured value was proved to be true by using the Suzuki–Fisher circular theorem [16], via a computation that can be found in [17]. The gap, q=3, was only recently filled by Beffara and Duminil-Copin in [18].

The free energy of the two-dimensional Ising model was explicitly computed by Onsager [15]. The Potts model for q=2 is equivalent to the Ising model—the only difference is that $\Phi^{\mathrm{Potts}}=\left(\Phi^{\mathrm{Ising}}+1\right)/2$. Taking this into account, and using the exact result by Onsager, we obtain
(12)$$\;-\beta \;f\left(\beta \right)=-\beta +\frac{\log(2)}{2}+\frac{1}{2\pi }\int_{0}^{\pi }\log\left(\cosh^{2}\left(\beta \right)+\frac{\sqrt{1+\kappa^{2}-2\kappa \;\cos\left(2\phi \right)}}{\kappa }\right)\;d\phi ,$$
where $\kappa =1/\sinh^{2}\left(\beta \right)$.

As mentioned above, the Potts model satisfies Hypothesis 1. It can be easily verified that the SFT resulting after a coding reducing the interaction to one-letter is strongly irreducible. This is a direct consequence of the fact that the original Potts model is defined on a full shift. Hence, we can construct the corresponding family of SFTs defined in the previous section. Theorems 1 and 2 ensure the following.

**Corollary** **1.**
*For each q≥2 and $N>\ell_{q}:=\sqrt[{4}]{\sqrt{q}+1}$, there is a strongly irreducible two-dimensional SFT on an alphabet of size $q\times {\left(1+\left(q-1\right)N\right)}^{4}$, defined by a collection of admissible patches on $\Lambda_{\mathrm{cross}}$, having exactly q different measures of maximal entropy, i.e., such that $\mathrm{Max}(X_{N})$ is equivalent to the standard (q−1)-simplex of probability vectors in dimension q.*


**Proof.** For q≥2, let *X* be the two-dimensional SFT on the alphabet ${\left\{0,1,\ldots ,q-1\right\}}^{\Lambda_{\mathrm{cross}}}$ codifying ${\left\{0,1,\ldots ,q-1\right\}}^{Z^{2}}$ by overlapping patches, and let Φ be the one-letter interaction induced by the Potts interaction using this codification. Fix *N* and consider the strongly irreducible SFT on the alphabet $A_{N}={⋃}_{s=0}^{4}\left(A_{s}\times \left\{0,1,\ldots ,N_{s}-1\right\}\right)$, as prescribed in Section 3. As stated in Theorem 2, the critical inverse temperature of the Potts model is $\beta_{c}\left(q\right)=\log\left(\sqrt{q}+1\right)/2$. According to Theorem 1, an inverse temperature $\beta_{N}:=\log\left(N\right)/\epsilon_{0}$ corresponds to each splitting multiplicity *N*; hence,
$$\beta_{N}>\beta_{c}\left(q\right)\;\mathrm{if}\;\mathrm{and}\;\mathrm{only}\;\mathrm{if}\;N>\ell_{q}:=\sqrt[{4}]{\log(\sqrt{q}+1)}.$$ On the other hand, Theorem 2 ensures that above $\beta_{c}\left(q\right)$, the simplex of equilibrium states, $E_{\Phi }\left(\beta_{N}\right)$, is equivalent to the standard (q−1)-simplex of probability vectors in dimension *q*. Therefore, from Theorem 1, we obtain that for each $N>\ell_{q}$, $\mathrm{Max}(X_{N})$ has exactly *q* different ergodic measures. Finally, it can be easily verified that
$$|A_{N}|=\sum_{s=0}^{4}N^{4-s}\left|A_{s}\right|=\sum_{s=0}^{4}N^{4-s}\;q\left(\begin{array}{c} 4\\ s \end{array}\right)q{\left(q-1\right)}^{4-s}=q\times {\left(1+\left(q-1\right)N\right)}^{4}.$$□

For the SFT $X_{N}$ corresponding to the *q*-colored Potts model at inverse temperature $\beta_{N}$, to admit q+1 different ergodic measures of maximal entropy, we need $N\in \{\ell_{q}:\;q\geq 5\}\cap N$. For those integers, $\mathrm{Max}(X_{N})$ is spanned by *q* ergodic measures, each one giving preference to each one of the colors k=0,1,…,q−1, and an extra ergodic measure for which all colors appear with the same proportion. The smallest system of this kind is obtained with q=225 and N=2.

For the family of strongly irreducible SFTs corresponding to the Ising model (Potts with q=2), we have $\ell_{2}=\sqrt[{4}]{\sqrt{2}+1}\approx 1.246504703$. Hence, while $X_{1}$ is intrinsically ergodic, $\mathrm{Max}(X_{N})$ has two extrema for each N≥2. In this way, the construction developed in Section 3 gives us a strongly irreducible SFT on an alphabet of size $2\times 3^{4}$ symbols with exactly two different ergodic measures. In general, the correspondence established by Theorem 1, applied to the Potts model, gives us a strongly irreducible two-dimensional SFT on an alphabet of size $q\times {\left(1+\left(q-1\right)\left\lceil \sqrt[{4}]{\sqrt{q}+1}\;\right\rceil \right)}^{4}$, defined by patches on the volume $\Lambda_{\mathrm{cross}}$. This is of course not the smallest cardinality required. A clever coding allows us to find alphabets of smaller cardinality achieving the correspondence. In Appendix A, we explicitly give a strongly irreducible SFT on the alphabet $\left\{0,1\right\}\times {\left\{0,1,2\right\}}^{2}$, admitting exactly two different ergodic measures.

In the case of q=2, which corresponds to the Ising model, $X_{N}$ is defined on an alphabet of size $2\times {\left(N+1\right)}^{4}$, by a collection of admissible patches on $\Lambda_{\mathrm{cross}}$. Therefore, the trivial upper bound for the topological entropy is in this case
$$h_{\mathrm{top}}\left(X_{N}\right)\leq 4\log\left(N\right)+\log\left(2\right),$$
but since the overlapping condition has to be respected, a finer upper bound can be obtained by taking into account horizontal and vertical adjacency restrictions. Indeed, in an admissible configuration *x*, the symbols $\pi_{c}\left(x_{o}\right)$ and $\pi_{c}\left(x_{e_{1}}\right)$, when considered as patches in ${\left\{0,1\right\}}^{\Lambda_{\mathrm{cross}}}$, have to share two letters. Therefore, given $\pi_{c}\left(x_{o}\right)$, the patch $\pi_{c}\left(x_{e_{1}}\right)$ has only three free positions at sites $e_{1},e_{2}$ and $-e_{2}$. If we suppose that the overlapping sites of patches $\pi_{c}\left(x_{o}\right)$ and $\pi_{c}\left(x_{e_{1}}\right)$ have the same letter, then according to which of the three free sites of the patch $\pi_{c}\left(x_{e_{1}}\right)$ are equal to the central site, we obtain
$$D_{N}=N^{4}+\left(\begin{array}{c} 3\\ 1 \end{array}\right)\;N^{3}+\left(\begin{array}{c} 3\\ 1 \end{array}\right)\;N^{2}+\left(\begin{array}{c} 3\\ 1 \end{array}\right)\;N=N{\left(N+1\right)}^{3}$$
possibilities for $x_{e_{1}}$. If, on the contrary, we suppose that the overlapping sites of patches $\pi_{c}\left(x_{o}\right)$ and $\pi_{c}\left(x_{e_{1}}\right)$ have different letters, then by a similar counting, we obtain $D_{N}/N$ different possibilities for $x_{e_{1}}$. The same kind of restriction applies for the vertical adjacency. From this, we readily obtain the upper bound
$$h_{\mathrm{top}}\left(X_{N}\right)\leq \lim_{n\to \infty }\frac{|A_{N}|\times D_{N}^{n-1}\;{\left(D_{N}^{n-1}\right)}^{n}}{n^{2}}=3\;\log\left(N+1\right).$$ Taking into account the fact that the Helmholtz free energy of the Potts model is known, then Proposition 1 allows us to exactly compute the topological entropy. We have the following.

**Corollary** **2.**
*The two-dimensional SFT $X_{N}$, corresponding to the Potts model via Theorem 1 with q=2 at inverse temperature $\beta_{N}=2\log\left(N\right)$, has topological entropy*

$$h_{\mathrm{top}}\left(X_{N}\right)=2\;\log\left(N\right)+\frac{\log(2)}{2}+\frac{1}{2\pi }\int_{0}^{\pi }\log\left({\left(\frac{N^{4}+1}{2\;N^{2}}\right)}^{2}+\frac{\sqrt{1+\kappa_{N}^{2}-2\kappa_{N}\;\cos\left(2\phi \right)}}{\kappa_{N}}\right)\;d\phi ,$$

*where $\kappa_{N}:={\left(2N^{2}/\left(N^{2}-1\right)\right)}^{2}$.*


This result directly follows from Onsager’s result and Proposition 1, by taking into account that the correspondence at inverse temperature $\beta_{N}$ gives $\kappa =\kappa_{N}$ in Equation (Equation 12).

## 5. The SFT–SMM Correspondence for the Six-Vertex Models

Besides the Potts model, there are a few SMMs for which the Helmholtz free energy is exactly known. Among them, we have the ice-type models or six-vertex models, introduced to model crystals with hydrogen bonds, such as ice crystal (see [19] and references therein, and [20] for a very recent account). Those models can be thought of as SMMs with support on a two-dimensional subshift of finite type *X* with entries in the alphabet A={ne,sw,se,nw,oi,io}. The symbols of A represent arrow configurations around a central node as indicated in the figure below.



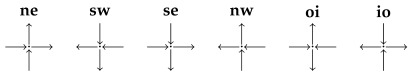



The collection of admissible patches $Ł\subset A^{\Lambda_{\mathrm{cross}}}$ determining *X* is built by the following rule: two symbols cannot occupy adjacent sites unless the arrow configurations they represent are such that the head of one arrow matches the tail of the neighboring arrow. The SFT is supplied with a one-symbol interaction giving rise to several submodels depending on the relative values of this interaction. All those submodels can fit into the framework of Theorem 1 by considering interactions that are an integer multiple of a given value. The first one of the submodels is known as the “Ice model”, for which the interaction is constant, and therefore it cannot produce a phase transition. The SFT obtained by means of the SFT–SMM correspondence is in this case the original SFT whose topological entropy was computed by E. Lieb in [21], giving
$$h_{\mathrm{top}}\left(X\right)=\frac{3}{2}\;\log\left(\frac{4}{3}\right).$$ This SFT is transitive but not strongly irreducible, and although it is not formally proved, it is expected that the system is not intrinsically ergodic. There are several results suggesting the coexistence of several measures of maximal entropy, in particular, those concerning the effect of the boundary conditions on the convergence of several indicators (see [22] for instance).

Two other important submodels are the KDP model and the Rys F model, which are described in detail in [19] (see [20] as well). They are defined on *X* by a one-letter interaction of the kind
(13)$$\Phi_{o}\left(a\right)=\left\{\begin{array}{cc} \epsilon_{0} & \;\mathrm{if}\;a\in \{\mathrm{ne},\mathrm{sw}\},\\ \epsilon_{1} & \;\mathrm{if}\;a\in \{\mathrm{se},\mathrm{nw}\},\\ \epsilon_{2} & \;\mathrm{if}\;a\in \{\mathrm{oi},\mathrm{io}\}, \end{array}\right.$$
with $\left\{\epsilon_{0},\epsilon_{1},\epsilon_{2}\right\}=\left\{0,1\right\}$. They therefore satisfy hypothesis H with S={0,1}. The KDP model corresponds to the choice $\epsilon_{0}=0<\epsilon_{1}=\epsilon_{2}=1$, and its behavior with respect to β displays two distinct regimes: high temperature when β<log(2) and low temperature where β≥log(2). At low temperature, the system admits two different ergodic equilibrium states supported by the two homogeneous configurations **ne**${}^{Z^{2}}$ and **sw**${}^{Z^{2}}$. The Helmholtz free energy of the KDP model is given by
$$-\beta \;f\left(\beta \right)=\left\{\begin{array}{cc} -\beta +\int_{R}\frac{1}{2x}\frac{\sinh(6\;\zeta \;x)}{\cosh(\zeta \;x)}\frac{\sinh((\pi -\zeta )\;x)}{\sinh(\pi \;x)}\;dx & \;\mathrm{if}\;\beta <\log(2)\\ -\beta & \;\mathrm{if}\;\beta \geq \log(2), \end{array}\right.$$
with $\cos\left(\zeta \right)=-e^{\beta }/2$. The SFT–SMM correspondence gives, for each N∈N, a transitive two-dimensional SFT $X_{N}$, on the alphabet $A_{N}=\left(\left\{\mathrm{ne},\mathrm{sw}\right\}\times \left\{0,1,\ldots ,N-1\right\}\right)⋃\left\{\mathrm{se},\mathrm{nw},\mathrm{oi},\mathrm{io}\right\}$. The topological entropy of each one of those SFTs can be exactly computed. Indeed, applying Proposition 1 to the KDP model at inverse temperature $\beta_{N}=\log\left(N\right)$, we obtain
$$h_{\mathrm{top}}\left(X_{N}\right)=\left\{\begin{array}{cc} 2/3\;\log\left(4/3\right) & \;\mathrm{if}\;N=1,\\ \log(N) & \;\mathrm{if}\;N\geq \log(2). \end{array}\right.$$
which is not as interesting as expected.

With $\epsilon_{0}=\epsilon_{1}=1>\epsilon_{2}=0$ in Equation (Equation 13), we obtain the Rys F model. Once again, its behavior with respect to β displays two regimes: disordered for β<log(2), and ordered for β≥log(2). At sufficiently low temperature, the system has a unique ergodic equilibrium state associated to periodic configurations alternating the symbols io and oi. The Helmholtz free energy for this model (which can be found in [20]) is
$$-\beta \;f\left(\beta \right)=\left\{\begin{array}{cc} -\beta +\int_{R}\frac{1}{2x}\frac{\sinh(2\left(\pi -\theta_{1}\right)\zeta_{1}x/\pi )}{\cosh(\zeta_{1}\;x)}\frac{\sinh(\left(\pi -\zeta_{1}\right)\;x)}{\sinh(\pi \;x)}\;dx & \;\mathrm{if}\;\beta <\log(2),\\ -\log\left(2\right)+\int_{R}\frac{e^{-|x|}}{2x}\frac{\sinh(x)}{\cosh(x)}dx\; & \;\mathrm{it}\;\beta =\log(2),\\ -\beta +\frac{\zeta_{3}\theta_{3}}{\pi }+\sum_{n=1}^{\infty }\frac{e^{-n\zeta_{3}}}{n}\frac{\sinh(2n\zeta_{3}\theta_{3}/\pi )}{\cosh(n\zeta_{3})} & \;\mathrm{if}\;\beta >\log(2). \end{array}\right.$$
with $\cos\left(\zeta_{1}\right)=e^{2\beta }/2-1$, $\sin\left(\theta_{1}\zeta_{1}/\pi \right)=\sqrt{1-e^{2\beta }/4}$, $\cosh\left(\zeta_{3}\right)=e^{2\beta }/2-1$ and $\sinh\left(\theta_{3}\zeta_{3}/\pi \right)=\sqrt{e^{2\beta }/4-1}$. The SFT–SMM correspondence gives, for each N∈N, a transitive two-dimensional SFT $X_{N}$, on the alphabet $A_{N}=\left(\left\{\mathrm{oi},\mathrm{io}\right\}\times \left\{0,1,\ldots ,N-1\right\}\right)⋃\left\{\mathrm{ne},\mathrm{sw},\mathrm{se},\mathrm{nw}\right\}$, for which the topological entropy can be exactly computed. Proposition 1 gives in this case
$$h_{\mathrm{top}}\left(X_{N}\right)=\left\{\begin{array}{cc} \frac{2}{3}\;\log\left(\frac{4}{3}\right) & \;\mathrm{if}\;N=1,\\ 2\;\log\left(2\;\frac{\Gamma (5/4)}{\Gamma (3/4)}\right) & \;\mathrm{it}\;N=2,\\ \mathrm{asinh}\left(\sqrt{\frac{N^{2}}{4}-1}\right)+\sum_{n=1}^{\infty }\frac{e^{-n\;\mathrm{acosh}(N^{2}/2-1)}}{n}\frac{\sinh(2n\;\mathrm{asinh}\left(\sqrt{N^{2}/4-1}\right))}{\cosh(n\;\mathrm{acosh}\left(N^{2}/2-1\right))} & \;\mathrm{if}\;N>2, \end{array}\right.$$
which turns out to be a little more interesting.

## 6. Final Remarks

We have seen how statistical mechanics models undergoing a phase transition can be used, by applying the SFT–SMM correspondence, to obtain strongly irreducible subshifts of finite type admitting several measures of maximal entropy. We have focused on SMM for which we have a precise description of the phase diagram, the family of Potts models, which allows us to explicitly obtain, for each q≥2, a two-dimensional SFT having a simplex of measures of maximal entropy equivalent to the standard (q−1)-simplex of probability vectors. Furthermore, the detailed description of the pure phases for the Potts model tells us how the ergodic measures of maximal entropy of these SFTs behave. For instance, we know that for each color of the alphabet, there exists a measure of maximal entropy for which the typical configuration contains an unbounded connected region of symbols of that color, surrounding bonded patches of symbols of the complementary colors. The rigorous study of the Potts model (in particular, the case q=2) gives, through the correspondence, examples of strongly irreducible SFTs for which characteristics such as the speed of decay of correlations, the shape of bounded regions of complementary colors (the Wulff shape), and the distribution of the size of the islands of those colors are precisely known. Besides the references already cited, a relatively recent account can be found in [23], where Martirosian’s Theorem is revisited. An even more recent and didactic review, concerning the Potts model and related subjects, can be found in [24].

The other important application of the SFT–SMM correspondence that we have illustrated concerns the construction of SFTs for which the topological entropy can be explicitly computed. Besides the three families of SFTs we have considered, which correspond to the Ising model and some instances of the six-vertex model, there are other vertex-type SMMs for which the Helmholtz free energy is known. They include other instances of the six-vertex model as well as eight-vertex models (see [19] for instance). Those models would provide other families of SFTs for which the topological entropy can be computed.

## Data Availability

Data sharing not applicable.

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
