# Peer review of "On the Correspondence between Subshifts of Finite Type and Statistical Mechanics Models"

_entropy, 2022, doi:10.3390/e24121772_

Round 1

Reviewer 1 Report

Dear Editor,

I have reviewed the article " On the correspondence between Subshifts of Finite Type and Statistical Mechanics Models   "with interest. This work is the extension of the classical work cited as ([18]: O. H¨aggstr¨om. “On the relation between finite range potentials and subshifts of finite type”, Probability Theory and 434 Related Fields (1995) 101, 469–478.)   with a low similarity index of 15%. A version of this article is available on ARXIV since April 2021. I recommend a minor revision (corrections to minor methodological errors and text editing) along with my following observations

1: The title is appropriate 

2: Abstract is written clearly but can be improved it to attract more readers in an aspiring way (by adding one or two lines in a more general context of the subject). It will have an impact on the citations of the article. 

 3: A discussion about the limitation of work in comparison with the (simplex of measures of maximal entropy) may also be added in the introduction section 1. However, the references at the end of this manuscript are not in mdpi format.  I suggest adding a general discussion (motivation) paragraph in the introduction section by adding related references to attract a large audience e.g (or any other references can be chosen). 

100 Years of the (Critical) Ising Model on the Hypercubic Lattice (mathunion.org)

Glasses and aging: A Statistical Mechanics Perspective (inria.fr)

4: The results are presented in a good way in light of the theoretical background and lead to a detailed summary and future directions. I hope this manuscript will not only be a good (classical) contribution to this journal but also to the literature. 

5. The article should be proofread for typos and minor English spell check.

All the best

Author Response

1: The title is appropriate 

2: Abstract is written clearly but can be improved it to attract more readers in an aspiring way (by adding one or two lines in a more general context of the subject). It will have an impact on the citations of the article. 

------- We added a few words to the abstract, situating the problem in a general context.

 3: A discussion about the limitation of work in comparison with the (simplex of measures of maximal entropy) may also be added in the introduction section 1. However, the references at the end of this manuscript are not in mdpi format.  I suggest adding a general discussion (motivation) paragraph in the introduction section by adding related references to attract a large audience e.g (or any other references can be chosen). 

100 Years of the (Critical) Ising Model on the Hypercubic Lattice (mathunion.org)

Glasses and aging: A Statistical Mechanics Perspective (inria.fr)

--------  We put all references in the required format. Nevertheless, we did not add any further discussions or comments in the introduction, since it would require a substantial time to do it correctly.

4: The results are presented in a good way in light of the theoretical background and lead to a detailed summary and future directions. I hope this manuscript will not only be a good (classical) contribution to this journal but also to the literature. 

5. The article should be proofread for typos and minor English spell check.

---------  The paper was revised by a proof reading expert. We followed all his corrections and most of his suggestions.

Reviewer 2 Report

In this work the authors proposed a simplified version of correspondence between equilibrium statistical mechanics and symbolic dynamics. The proposed construction makes explicit the correspondence between
 the parameterization of the family of SFTs and the inverse temperature in the corresponding SMM. This helps them to point out further applications of statistical mechanics results to symbolic dynamics. Also, the authors used Potts model and the six-vertex model of statistical mechanics to illustrate this correspondence. The paper is nicely written with enough background to understand the results. The obtained results are a good advancement in statistical mechanics models. Therefore I would like to recommend the paper for publication. Few of my suggestions are

·         Generally I ask the authors to read the paper carefully for grammatical mistakes and typos.

·         There is no need to write in bracket (SFT) wherever you write subshift of finite type.

·         In section 2, there is no need to give numbers like 2.1, 2.2 and 2.3. Instead the authors can made subsections and give them a suitable title.

·         Similarly in other section, it is better to remove the numbers.

Author Response

Generally I ask the authors to read the paper carefully for grammatical mistakes and typos.

 ------ The paper was revised by a proof reading expert. We made the indicated corrections and followed the pertinent indications.

There is no need to write in bracket (SFT) wherever you write subshift of finite type.

------- We have eliminated all the appearances of (SFT).

In section 2, there is no need to give numbers like 2.1, 2.2 and 2.3. Instead the authors can made subsections and give them a suitable title.

Similarly in other section, it is better to remove the numbers.

-------- We have eliminated those numberings.